# Toxicity and Clinical Results after Proton Therapy for Pediatric Medulloblastoma: A Multi-Centric Retrospective Study

**DOI:** 10.3390/cancers14112747

**Published:** 2022-06-01

**Authors:** Alessandro Ruggi, Fraia Melchionda, Iacopo Sardi, Rossana Pavone, Linda Meneghello, Lidija Kitanovski, Lorna Zadravec Zaletel, Paolo Farace, Mino Zucchelli, Mirko Scagnet, Francesco Toni, Roberto Righetto, Marco Cianchetti, Arcangelo Prete, Daniela Greto, Silvia Cammelli, Alessio Giuseppe Morganti, Barbara Rombi

**Affiliations:** 1Specialty School of Paediatrics-Alma Mater Studiorum, Università di Bologna, 40138 Bologna, Italy; alessandro.ruggi@studio.unibo.it; 2Pediatric Onco-Hematology, IRCCS Sant’Orsola SSD, University Hospital of Bologna, 40138 Bologna, Italy; fraia.melchionda@aosp.bo.it (F.M.); arcangelo.prete@aosp.bo.it (A.P.); 3Neuro-Oncology Unit, Department of Pediatric Oncology, Meyer Children’s Hospital, 50139 Florence, Italy; iacopo.sardi@meyer.it (I.S.); rossana.pavone@meyer.it (R.P.); 4Pediatric Onco-Hematology Service, Pediatric Unit, Santa Chiara Hospital, 38123 Trento, Italy; linda.meneghello@apss.tn.it; 5Department of Oncology and Haematology, University Children’s Hospital, University Medical Centre Ljubljana, 1000 Ljubljana, Slovenia; lidija.kitanovski@kclj.si; 6Division of Radiotherapy, Institute of Oncology, 1000 Ljubljana, Slovenia; lzaletel@onko-i.si; 7Proton Therapy Unit, Santa Chiara Hospital, Azienda Provinciale per i Servizi Sanitari (APSS), 38123 Trento, Italy; paolo.farace@apss.tn.it (P.F.); roberto.righetto@apss.tn.it (R.R.); marco.cianchetti@apss.tn.it (M.C.); 8Pediatric Neurosurgery, Institute of Neurological Science, IRCCS Bellaria Hospital, 40139 Bologna, Italy; minoz@inwind.it; 9Department of Neurosurgery, Meyer Children’s Hospital, 50139 Florence, Italy; mirko.scagnet@meyer.it; 10Neuroradiology Unit, IRCCS Istituto delle Scienze Neurologiche di Bologna, 40139 Bologna, Italy; f.toni@isnb.it; 11Azienda Ospedaliero-Universitaria Careggi, 50134 Florence, Italy; daniela.greto@unifi.it; 12Radiation Oncology, IRCCS Azienda Ospedaliero-Universitaria di Bologna, 40138 Bologna, Italy; silvia.cammelli2@unibo.it (S.C.); alessio.morganti2@unibo.it (A.G.M.); 13Department of Experimental, Diagnostic and Specialty Medicine-DIMES, Alma Mater Studiorum, University of Bologna, 40138 Bologna, Italy

**Keywords:** pediatric brain tumors, proton therapy, medulloblastoma, radiation, toxicity

## Abstract

**Simple Summary:**

Medulloblastoma is the most common malignant brain tumor in children. Treatment is effective, but survivors often develop long-term sequelae that impact their quality of life. Proton therapy has similar efficacy as traditional radiotherapy but might achieve lower toxicity. We present our retrospective results on 43 children treated with active scanning proton therapy for medulloblastoma. This case series confirms that protons for medulloblastoma are associated with a favorable toxicity profile. Results on late effects and survival, although preliminary, are encouraging.

**Abstract:**

Medulloblastoma is the most common malignant brain tumor in children. Even if current treatment dramatically improves the prognosis, survivors often develop long-term treatment-related sequelae. The current radiotherapy standard for medulloblastoma is craniospinal irradiation with a boost to the primary tumor site and to any metastatic sites. Proton therapy (PT) has similar efficacy compared to traditional photon-based radiotherapy but might achieve lower toxicity rates. We report on our multi-centric experience with 43 children with medulloblastoma (median age at diagnosis 8.7 years, IQR 6.6, M/F 23/20; 26 high-risk, 14 standard-risk, 3 ex-infant), who received active scanning PT between 2015 and 2021, with a focus on PT-related acute-subacute toxicity, as well as some preliminary data on late toxicity. Most acute toxicities were mild and manageable with supportive therapy. Hematological toxicity was limited, even among HR patients who underwent hematopoietic stem-cell transplantation before PT. Preliminary data on late sequelae were also encouraging, although a longer follow-up is needed.

## 1. Introduction

Medulloblastoma (MB) is one of the most common brain tumors in the pediatric age group, most cases being diagnosed between 6 and 8 years of age [1]. Even though with current treatment options prognosis has improved, MB survivors often develop long-term physical and psychological sequelae because of surgery, radiotherapy, and chemotherapy, with negative impact on their quality of life [2,3,4,5,6]. Cognitive deficits are common in long-term survivors, and many cannot live a fully independent life [7,8]. There is a need to identify new and less toxic therapeutic strategies [1,7,9].

Standard-of-care for MB includes surgical resection, cytotoxic chemotherapy, and, in non-infants, craniospinal irradiation (CSI) plus a boost to the posterior fossa primary tumor site and to any metastasis seen on the latest preirradiation MRI, with variable doses for standard-risk (SR) and high-risk (HR) patients. In terms of outcome, 5-to-10-year overall survival for SR is approximately 80% while the 5-year rate is around 70% for HR disease [1,9,10,11,12].

The irradiation of the entire CNS is the major source of morbidity in long-term MB survivors, with toxicity being directly proportional to radiation dose and treated volume [13,14,15]; however, it is necessary since MB has a high tendency for leptomeningeal spread, which is the main cause of death from this disease [9]. Usually, for children under 3 years, radiation-sparing protocols are preferable due to the severity of CSI-induced long-term sequelae in this population [12,16].

MB is the pediatric brain tumor most commonly treated with proton therapy (PT) [17,18,19,20]. This radiation modality has similar efficacy in terms of survival and local control (LC) compared to traditional photon-based radiotherapy, but it deposits a smaller radiation dose to healthy tissues, which may result in reduced acute and late toxicities. PT may improve neurocognitive outcomes [21,22] and reduce second malignancies [23] and neuroendocrine toxicity [24], while its benefit in terms of ototoxicity is still being evaluated [25,26]. Moreover, due to the negligible dose delivered to the organs anterior to the vertebral bodies with proton-based CSI, this could lead to lower rates of cardiac [27] and gonadal toxicity [28] as well. The lens can also be greatly spared during CSI with protons [29]. PT is well-tolerated in the acute setting, with the potential of reducing hematological and gastrointestinal toxicities [29,30,31].

The aim of the present study is to present our retrospective, multi-institutional results on the use of PT for the treatment of both SR and HR childhood MB, with a focus on its acute and subacute toxicities, including hematological side effects. We will also present preliminary data on late effects, disease control, and survival.

## 2. Materials and Methods

This multi-centric retrospective study was approved by the Institutional Review Board at all institutions. Inclusion criteria were as follows: (1) histologically confirmed MB diagnosis, (2) age ≤18 years at diagnosis, and (3) no history of prior radiation.

All patients were treated according to their referring institution’s protocols, based on the best available evidence and current treatment standards for MB. The quality and timing of imaging and diagnostic procedure, the extent of surgery, and response evaluation were defined according to the SIOP Brain Tumor Subcommittee criteria [32]. Gross total resection (GTR) of the tumor was attempted in all patients, whenever feasible. Date of diagnosis was defined as the date of the first MRI/CT showing a CNS mass. At the time of diagnosis, all patients received contrast-enhanced MRI and/or CT scan. Extent of surgery was evaluated based on combined surgical and radiological judgment.

Tumor staging was performed with brain MRI and/or CT before and after surgery, as well as spine MRI and cerebral-spinal fluid (CSF) cytology. Minimal residual disease (<1.5 cm²) and no evidence of metastasis (M0), without *MYC* amplification and/or anaplastic histology defined SR disease. Visible metastatic disease or positive CSF cytology, as well as residual disease after surgery of 1.5 cm² or more, the presence of *MYC* amplification, and/or large cell/anaplastic histology, were classified as HR. All pathology for the Italian patients was reviewed by the Italian central institution for brain tumors.

SR patients (14 cases) received upfront PT after surgery, followed by adjuvant chemotherapy based on cisplatin, CCNU, and vincristine. Five SR patients also received concurrent, single-agent vincristine once a week during irradiation. HR patients (26 cases) received upfront intensive induction chemotherapy, followed by PT. Eleven HR patients also underwent autologous HSCT before PT. Twelve HR patients received concurrent, single-agent vinorelbine once every two weeks during irradiation, while one HR patient received concurrent temozolomide.

Forty patients received PT as part of their first-line treatment. We included in the cohort three ex-infant MB patients who were treated with PT as part of their second-line treatment after a first progression: they had not received any prior radiation and they were older than 3 years at PT.

All patients in this study received PT at the Trento Proton Therapy Center. PT was delivered with active pencil-beam scanning (PBS). Technical details on the treatment procedure have already been described elsewhere [33,34].

Craniospinal target volume delineation for high-precision radiotherapy followed the guidelines published by the SIOPE Brain Tumor Radiotherapy Group [35]. All patients had a CT scan with 2 mm slices through the entire cranium and the whole spinal region to define the clinical target volume (CTV) for CSI. The cranial CTV included the whole brain, the cribriform plate, skull base foramina and canals, the pituitary lodge, and the optic nerves.

For skeletally mature patients, the spinal CTV included the subarachnoid space and spinal nerve roots, while for skeletally immature children, spinal CTV also included the vertebral bodies, to avoid bone deformity. The inferior border of spinal CTV was identified as lower limit of the thecal sac on the latest preirradiation sagittal spine MRI, generally at S2–S3 vertebral level. The planning target volume (PTV) was created as a 3–5 mm uniform expansion of the CTV. In addition, the following organs at risk (OARs) were outlined: lens, retinae, eyes, optic nerves, hypothalamus, pituitary gland, chiasm, posterior optic tracts, cochleae, brainstem, temporal lobes, hippocampi, spinal cord, thyroid gland, larynx, esophagus, heart, lungs, bowel, spleen, liver, kidneys, and the gonads.

For SR patients, the prescribed dose was 23.4 GyRBE in 13 fractions (1.8 GyRBE/day). For HR, CSI dose was 36 GyRBE delivered in 20 daily fractions (1.8 GyRBE each). For skeletally immature HR patients, CSI was delivered in two consecutive phases: in the first phase (19.8 GyRBE) the vertebral bodies were included in the target, while in the second phase the remaining 16.2 GyRBE was administered with vertebral body sparing to mitigate hematological toxicity in the particularly vulnerable cohort of HR patients. An additional boost was delivered to the primary tumor bed (up to 54–55.8 GyRBE) and to any visible metastatic sites.

Follow-up time was measured from the first day of PT. Disease status was monitored at regular intervals during and after PT based on imaging, laboratory tests, and clinical evaluations. During PT, weekly complete blood counts (CBCs) were performed to monitor hematological toxicity. Audiological, ophthalmological, and endocrinological follow-up was planned for all patients. Hormone status was evaluated at baseline, then at 3 and 6 months after PT, and later yearly. Ototoxicity was evaluated at 6 months and then yearly. Neuropsychological status was assessed at baseline and then yearly after irradiation: however, neurocognitive toxicity was evaluated systematically only in a minority of patients and will be the object of a separate analysis. We used the Common Terminology Criteria for Adverse Events (CTCAE), version 4.03 (2010), to score toxicity. Adverse effects occurring up to 90 days from the first day of PT were defined as acute, between 90 days and 6 months as subacute, and after 6 months as late. We recorded any acute and late adverse effects that could be partly related to the treatment.

### Statistical Analysis

All data were collected using a Microsoft Excel database. We then conducted a descriptive analysis of central tendency and dispersion measures (mean and/or median with associated standard deviation—SD—or interquartile range—IQR, respectively). Nominal and ordinal variables are presented as number and/or percentage of subjects.

Weekly mean hematological parameters (leukocytes, hemoglobin, platelets) during PT were compared between patients receiving high-dose CSI and patients receiving standard-dose CSI using Student’s *t*-test for independent samples, or Welch’s adaptation of the *t*-test in the case of unequal variances. Statistical significance level was set to be *p* < 0.05 for all tests.

Data analysis was conducted using IBM SPSS Statistics for Windows, Version 25.0., Armonk, NY: IBM Corp, USA, and MedCalc^®^ Statistical Software version 20.100 (MedCalc Software Ltd., Ostend, Belgium; https://www.medcalc.org (accessed on 2 April 2022); 2022).

## 3. Results

We report on our multi-centric experience of 43 children (23 males, 20 females) with MB treated with PT between 2015 and 2021, focusing on PT-related acute and subacute toxicity, with preliminary data on clinical results and late toxic side effects.

### 3.1. Population

The median age at diagnosis was 8.7 years [min–max: 1.8–18.6 years, IQR 6.6 years], while median age at PT was 8.9 years [3.0–19.0 years, IQR 6.8 years]; median follow-up was 26.0 months (2.0–67.0 months, IQR 30.5 months).

Fourteen (32.5%) had SR MB; twenty-six (60.5%) were considered HR. Lastly, three patients (7.0%) had an infant MB and thus were initially treated with a radiation-free protocol because of their young age; they received PT after three years of age for disease progression (“ex-infant” patients). Chang stages for HR patients were: 12 M0, 2 M1, 2 M2, and 10 M3. Two ex-infant patients were M0, one was M3.

Histologies were as follows: 31 classic, 3 desmoplastic, 7 large cell/anaplastic, 1 extensive nodularity, 1 NOS. As for available molecular subgroups, 29 were non-SHH/non-WNT, 8 SHH, 3 WNT. *MYC* amplification was present in 2 HR patients. In three cases, molecular analysis was not available.

All patients in our cohort received at least one surgical intervention before PT. Thirteen (50.0%) HR patients and five (35.7%) SR patients received concurrent chemotherapy during PT, in accordance with the treatment protocol used in their institution: vinorelbine once every two weeks for HR patients, vincristine once weekly for SR patients. Fourteen (32.6%) patients had received an autologous HSCT before PT.

The median duration of PT was 42 days [37–57 days, IQR 2 days].

Sedation for PT was required for children under five years of age and for those in which adequate compliance was not possible.

Table 1 and Table 2 provide a summary of patient characteristics.

### 3.2. Toxicity Analysis

For the PT toxicity analysis, we divided the patients into two groups, based on CSI dose: the high-dose group (36.0–39.6 GyRBE, *n* = 27) comprises 25 HR patients and 2 ex-infant patients; the standard-dose CSI group (23.4–25.2 GyRBE, *n* = 16) comprises all SR patients, one ex-infant, and one HR patient.

#### 3.2.1. Acute and Subacute (<6 Months) Toxicities

A summary of acute and subacute toxicities can be found in Table 3. No G4–G5 toxicities were recorded. The most common (>30% of patients) toxicities were radiation dermatitis (*n* = 28, 65.1%), pharyngeal mucositis (22, 51.2%), nausea (19, 44.1%), non-preexisting alopecia (17 out of 18 without preexisting alopecia, 94.4%), anorexia (16, 37.2%), and fatigue (15, 34.9%): these toxicities appeared usually during the first month of PT and were predominantly mild (G1 or G2), with few cases reaching G3. Herpes zoster infection or reactivation was relatively common (7 cases, 16.3%), without a clear correlation with HR or SR group, previous chemotherapy, or steroid therapy concurrent to PT. We observed one G3 case of anorexia 5 months after PT start, which resulted in a 20% body weight loss: the patient was receiving chemotherapy with CCNU, which was considered the main cause of the symptoms. In accordance with the treatment protocol, the CCNU dose was halved, and the patient received total parenteral nutrition at home, with gradual recovery. One patient developed a G3 posterior reversible encephalopathy (PRES) approximately 3.5 months after PT; the episode completely resolved spontaneously, and neuroradiological follow-up showed improvement of the lesions. This episode of PRES happened during consolidation therapy with vincristine and CDDP and we attributed this complication to vincristine rather than to PT, as it is a known effect of this drug [36].

#### 3.2.2. Hematological Toxicity

Figure 1a–c shows the trend of hematological values as clustered boxplots by week of treatment and by CSI dose. Table 4 shows the results of a comparison of weekly mean hematological values between patients who received high-dose CSI and standard-dose CSI. A significant difference in leukocyte count was found at weeks 1, 2, 3, and 5, with patients receiving high-dose CSI having lower counts. No significant differences in hemoglobin values were observed. Platelet count was significantly lower in patients receiving high-dose CSI at every timepoint. The nadir in leukocytes and platelets was observed around the third week (days 15–21) of PT for both groups, while hemoglobin values remained stable. However, nine (20.9%) patients received red blood cell transfusions during PT, in accordance with the treatment protocol of their referring institution, which prescribes that Hb levels must be maintained above 10 g/dL during radiation. These six patients were receiving concurrent chemotherapy (vinorelbine every two weeks for HR patients, vincristine once weekly for SR patients). In one case, concurrent chemotherapy could not be administered because of low Hb levels, but irradiation was not interrupted.

Two (4.7%) patients received G-CSF for leukopenia: they were both HR patients who had received HSCT prior to PT and were also receiving concurrent chemotherapy with vinorelbine during irradiation.

One HR patient, who had received HSCT prior to PT, developed anemia and thrombocytopenia 6 months after the start of PT, requiring one transfusion of platelets and one transfusion of red blood cells.

The significantly different treatment before PT of HR and SR patients represents, however, an important variable that should be taken into account for a proper evaluation of hematological toxicity.

#### 3.2.3. Preliminary Data on Neuroendocrine Toxicities

Data on neuroendocrine outcomes are still limited due to the short follow-up. Six patients out of forty-three (14.0%) began replacement therapy because of hormonal deficits that developed after PT (non-preexisting). Median latency for the start of replacement therapy after PT was 9.5 months (min–max: 3.0–25.0 months). Four patients required monotherapy, in three cases with thyroxine, in one case with hydrocortisone. Two patients are currently receiving multi-drug replacement therapy because of a clinical picture of panhypopituitarism developing 25 and 5 months after PT, respectively. One of them had received a PT boost on a pituitary metastasis to a total dose of 54 GyRBE, while the other five patients received a cumulative median dose in the pituitary/hypothalamic region of 41 GyRBE. Of the five patients who are receiving thyroxine, four have a central hypothyroidism. Table 5 contains a summary of these data.

#### 3.2.4. Preliminary Data on Ototoxicity

Audiological assessments included bilateral measurement by conventional and extended high-frequency audiometry (1, 2, 3, 4, 6, and 8 kHz audiogram). Hearing impairment grading was based on a threshold shift >20 dB at 8, 4, and 3 kHz in at least one ear or audiological indication for cochlear implant (see CTCAE v4.03 for details).

Seven patients (16.3%) developed hearing impairment at follow-up audiometry, in five cases bilateral, in two cases monolateral. Median latency of hearing impairment was 9 months after PT (min–max: 2–37 months). Both patients with monolateral impairment had a first-degree relative with hearing loss and one of them is also affected by Gorlin syndrome. Severity of hearing impairment was defined as G1 in 1 case (2.3%), G2 in 3 cases (7.0%), and G3 in 3 cases (7.0%). All patients received platin-based chemotherapy; 4 out of 7 had received it before radiation. Table 6 contains a summary of these data.

Among patients who developed ototoxicity, mean dose to the right cochlea was 36.0 GyRBE (min–max: 23–53 GyRBE). Mean dose to the left cochlea was 34.6 GyRBE (min–max: 26–42 GyRBE).

One patient already had G3 hearing loss at baseline, before PT. This patient had received platin-based chemotherapy and the hearing loss remained stable at follow-up audiograms 3 years after PT.

#### 3.2.5. Preliminary Data on Late Toxicities

The most serious late effect was one case of cytomegalovirus (CMV) encephalitis 6 months after PT, which, despite early treatment with ganciclovir, eventually caused a bilateral permanent amaurosis (G4).

One patient had a G1 stroke 21 months after PT that presented with short-lasting aphasia. MRI showed a small ischemic area in the left lenticular nucleus. The patient is currently receiving prophylactic therapy with acetylsalicylic acid.

One patient developed a G2 intracranial bleeding 24 months after PT, without neurological sequelae, but shortly after was found to have developed panhypopituitarism (a pituitary deficit was already present at diagnosis, but gradually worsened over time): this patient had received a boost on a pituitary metastasis.

Follow-up MRIs found signs of cavernous malformations in 10 (23.3%) patients using SWI sequences, which developed at least 6 months after PT. Nine cases were completely asymptomatic. In one case, the patient presented with disorientation and headache in the Pediatric Emergency Department, and neuroradiological exams found signs of a small bleeding that likely originated from a cavernoma. The episode resolved spontaneously, and the patient was only monitored clinically.

One patient also developed osteoporosis of the lumbar vertebrae, which might be related to a PT boost on a metastasis in the lumbar spine. He is asymptomatic, had no complications, and is currently receiving oral therapy with vitamin D and calcium.

There were no cardiac effects, pulmonary disorders, gastrointestinal effects, esophageal strictures, chest wall abnormalities, or cases of dry mouth.

Assessment of neurocognitive sequelae is not part of the present study, since a longer follow-up is needed for proper evaluation and will be the subject of a separate analysis.

The late toxicities that we recorded are shown in Table 7.

### 3.3. Preliminary Data on Disease Control and Survival

Median follow-up is 26.0 months [2.0–67.0 months, IQR 30.5 months] and 41 patients are alive at the latest follow-up.

Median follow-up for SR patients is 28.5 months [2.0–66.0 months, IQR 22.8 months] and they are all alive. Two SR patients had PD 27 and 25 months after PT.

Median follow-up for HR patients is 20.0 months [5.0–67.0 months, IQR 31.3 months]. Disease control is as follows: 20 CR, 3 SD, 3 PD. Disease progression happened at 12, 15, and 23 months after PT. Two HR patients died because of disease progression, 19 and 34 months after PT.

All three ex-infant MB patients are in CR and are alive.

No patients had extraneural failures. No patients developed second malignancies.

## 4. Discussion

As previously stated, the aim of the present study is to present retrospective results for the use of PT for MB, its acute and subacute toxicity, and preliminary data on late effects, disease control, and survival, given the relatively short follow-up duration.

MB is the most common pediatric brain tumor treated with PT [20]. Even if available data about clinical efficacy, safety, and toxicities of PT in comparison to photon therapy for MB are encouraging, logistical and financial aspects remain an obstacle for a wider use of this technique. Nevertheless, several authors [17,18,19] believe that it should be strongly considered for MB treatment in children.

In our cohort, all 14 SR patients are alive at a median follow-up of 28 months, with 2 SR patients experiencing progression at 27 and 25 months after PT. It should be noted that one of them started PT after 105 days (which is a negative prognostic factor) from gross total resection: this delay was caused by severe post-surgical complications. Of 26 HR patients, at a median follow-up of 20 months, 3 patients had disease progression (at 12, 15, and 23 months after PT), 3 patients have an SD, and 20 are in CR. Two HR patients died because of disease progression, at 19 and 34 months after PT. The systematic evaluation of survival and treatment response is not the aim of the present study: treatment protocols, although similar, were heterogeneous between institutions and median follow-up is still too short. In terms of outcome, recent studies show for SR MB a 5-year event-free survival of more than 80%, and up to 70% for HR MB [12]. A recent phase-2 single-arm study of pediatric medulloblastoma treated with protons showed a 5-year PFS of 85% and 70% for the SR and HR groups, respectively, while 5-year overall survival was 86% and 75% [26].

PT was well-tolerated in terms of acute toxic effects. The most common (>30% of cases) observed acute toxicities were dermatitis, pharyngeal mucositis, nausea, alopecia, anorexia, and fatigue, in line with the current literature [31]. While nausea and vomiting rates are similar between patients who receive protons and photons (likely because nausea is triggered by the irradiation of the area postrema and the dorsal vagal complex in the CNS), PT is associated with a lower incidence of diarrhea [30]; this might be a consequence of the lower dose delivered to the bowel when using protons. In our cohort, all patients received prophylactic anti-emetic treatment during PT: nineteen patients (44.2%) still developed nausea and/or vomiting, reaching G3 only in one case; only one patient developed mild diarrhea in the first weeks of PT.

Radiotherapy hinders regional immune response, facilitating herpes virus primary infection or reactivation. This may compromise the benefits of cancer treatment, hide toxicities, cause interruptions, or prolong the treatment course [37]. In our study, 7 patients (16.3%) had a mucocutaneous varicella-zoster virus infection in the proximity of the irradiated field, during or immediately after PT. In addition to the possible cutaneous involvement, cranial radiotherapy is also associated with an increased risk of herpes encephalitis or meningitis, which might be fatal [37]. At the first signs of herpes virus infection/reactivation, treatment with nucleoside analogues should be initiated, given that the therapeutic window is small. Furthermore, CMV, another herpesvirus, can cause life-threatening CNS infections during cancer treatment. In our study, one case of CMV encephalitis, occurring 6 months after PT irradiation, resulted in permanent bilateral amaurosis, despite treatment with ganciclovir. Clinicians should have a high index of suspicion for herpes virus infections during and after PT, as in any other form of chemo and radiation treatment, because these viruses can cause severe and possibly fatal adverse events and require early treatment.

Hematologic toxicity is to be expected with CSI, because of the radiation dose received by the bone marrow and secondary lymphoid organs. There is a risk of leukopenia, anemia, and thrombocytopenia [38,39,40,41], especially in younger patients and in patients who receive chemotherapy before radiation [40]. A recent study by Liu et al. showed that proton CSI leads to significant reductions in acute hematological toxicity compared to photon CSI, with counts being significantly higher in the proton cohort in weeks 3 to 6 of radiation therapy [38]; the study mainly included SR MB cases. Our findings are consistent with these observations. Trends in leukocyte and platelet counts show a nadir in the third week of radiation (days 15 to 21) for all patients, which is consistent with the results of Liu et al. [38]. In our analysis, patients who receive high-dose CSI (25 HR and 2 ex-infant patients) have significantly lower leukocyte and platelet counts compared to patients who receive standard-dose CSI (14 SR, 1 HR, and 1 ex-infant patient), with no significant difference in hemoglobin levels, both at the start of PT and at later timepoints during irradiation. The fact is that all HR patients except one had received adjuvant chemotherapy before undergoing PT and in many cases (11 out of 25) also an HSCT. Furthermore, all three ex-infant patients had undergone HSCT before PT. All these patients tend to have worse baseline counts; they receive a higher dose of radiation and are therefore especially prone to hematological complications. To mitigate this effect of higher-dose CSI, only for skeletally immature children, vertebral bodies are included in the target during the first phase of PT (19.8 GyRBE, 11 fractions) only: studies have shown, in fact, that hematological toxicity can be reduced by lowering the dose on vertebral bodies [42]. A comparison between patients treated with different vertebral body irradiation approaches based on skeletal maturity could provide further informative data on hematological toxicity. Nevertheless, such a comparison was not feasible in the present study due to the limited patient number in each subgroup. Our experience also shows that HR patients tolerated PT well, even with concurrent single-agent chemotherapy with vinorelbine in 12 of them, and with concurrent temozolomide in one case: none had to interrupt radiation due to major hematological adverse effects, although in one case it was necessary to interrupt concurrent chemotherapy and in two cases G-CSF was administered for leukopenia. Nine patients (20.9%) received RBC transfusions during PT, but only because their referring institution treatment protocols mandate Hb levels of more than 10.0 g/dL during radiation.

Late toxicities after CSI for MB are common and diverse [43]. Follow-up in our study is still too brief to be able to extensively report on late toxicities, but our preliminary results are comparable to other studies.

Both hormonal and growth deficits after CSI are frequent [44]. These deficits develop over the years after radiation and are correlated with the dose delivered to the hypothalamic-pituitary region, to other endocrine organs, and directly to the developing skeleton. A study by Eaton et al. [24] compared proton and photon radiotherapy for SR MB, finding a reduced incidence of some, but not all, endocrine deficits after PT, and a lower impact on height deficit. A phase II study by Yock et al. on the late effects of PT in pediatric MB found an incidence of any hormone deficit of 63% at 7 years from the start of PT [26]. Six (14.0%) patients in our cohort are receiving hormone replacement therapy, especially with thyroxine (five cases, four of which are because of central hypothyroidism). This figure is likely going to increase. The median follow-up of 26 months is still relatively short and both neuroendocrine deficits and the impact on growth of cancer treatment will be better evaluated in time. The incidence of hormone deficiency was highest among patients who received a hypothalamic/pituitary median dose >40 GyRBE (among patients who are receiving hormone replacement therapy, hypothalamic/pituitary median dose was 41.24 GyRBE), confirming similar results reported in other studies [45].

Ototoxicity is a known possible side effect of MB treatment: both platin-based chemotherapy and radiotherapy contribute to this, with higher cochlear D_mean_ being associated with higher risk and degree of hearing loss [46]. The benefits of PT in terms of reduction of ototoxicity are still unclear. A recent study by Paulino et al., for example, showed similar ototoxicity rates between patients with MB treated with photons vs. protons, even if the dose to the cochlea was lower in the proton cohort [25]. We recorded seven cases (16.3%) of hearing impairment that developed at follow-up audiograms after PT. Severity of hearing loss in our patients at the latest follow-up was: 1 G1 (2.3%), 3 G2 (7.0%), and 3 G3 (7.0%). Our median follow-up is still too short to draw definitive conclusions on ototoxicity. In the aforementioned phase II study by Yock et al., the cumulative incidence of ototoxicity at 5 years from PT was 16% [26]. According to Paulino et al., a mean cochlear dose of 37 Gy can be considered a threshold for the risk of higher grade (3–4) ototoxicity and, below this threshold, the role of platin-based chemotherapy becomes prevalent in determining hearing loss [25,46]. In general, incidence of hearing loss is low for cochlear doses <30 Gy and increases if the mean cochlear doses are of more than 40–45 Gy [47]. The patients who developed hearing loss in our cohort had received a mean cochlear dose of 36.0 GyRBE (min–max: 23–53 GyRBE) on the right and 34.6 GyRBE (min–max: 26–42 GyRBE) on the left ear; and in one patient who developed right ear G3 hearing loss, the right cochlear D_mean_ was 53 GyRBE.

In our study, we recorded 11 (1 subacute, 10 late; 25.6% of the whole population) cases of cavernomas and/or cerebral microbleeds at follow-up MRIs using SWI sequences. Ten (1 subacute and 9 late) of these appeared in patients receiving high-dose CSI. The possible development of these vascular changes after radiation for brain tumors and MB in particular is well-documented for photons [48,49,50,51,52,53,54]. In a recent study, prevalence of cerebral microbleeds was 90% after a follow-up of 13.5 years in pediatric brain tumor survivors who received radiotherapy [52], with longer follow-up and higher CSI dose correlating with higher lesion count. Cavernomas and cerebral microbleeds are associated with higher irradiated brain volume, higher radiation doses, younger age at irradiation, and longer follow-up [52,55,56], and have been associated with poorer neurocognitive function [57]. In our patients, they appeared in the tumor bed boost area or in the brain areas that received at least 36.0 GyRBE; this observation is consistent with the results of Kralik et al. and Roddy et al., who report that the majority of cerebral microbleeds and cavernomas appear in areas that receive more than 30 Gy [55,57]. Data on their appearance after PT are limited, but they may appear with a shorter latency and are not uncommon [55]. The best management strategy for these malformations and their natural history is not yet clear: in most cases, they remain asymptomatic and experts believe that a specific treatment should be initiated only for selected symptomatic cases [48,51].

We recorded no cardiac adverse effects, by contrast with Jakacki and colleagues who reported 31% incidence of pathologic Q waves on ECG after at least one year from radiation and impaired maximal cardiac index in 75% of patients treated with photon CSI [58]. Additionally, we recorded no chest wall abnormalities by contrast with a 10.4-timeshigher risk in the childhood cancer survivor cohort [59]. No patient in our study had a diagnosis of restrictive lung disease, which can occur in 24–50% of long-term survivors treated with electron and ^6o^Co photon irradiation [60,61]. We recorded no new cases of gastrointestinal toxic effects such as esophageal strictures or dry mouth that have been previously recorded in up to 44% of patients treated with photons [62]. New-onset seizures have been recorded in 8% of patients in other studies [63]; one patient (2.3%) in our study developed absence seizures. There were no second tumors yet, which compares favorably with the 2% reported at 5 years in a previous trial of patients with SR medulloblastoma [10]. We observed no cases of radiation necrosis, which is a potentially severe but rare complication of radiation. The incidence of radiation necrosis in published photon series is around 4–5% [64,65] and recent studies on PT report similar rates [26,66,67].

This study has some limitations. First, being a retrospective, uncontrolled, and not randomized study, the evidence it provides is of inferior quality and is subject to bias. However, it should be noted that some experts believe that randomized trials of proton and photon radiotherapy in children are ethically questionable and not feasible [68,69,70]. Moreover, as previously stated, there was some heterogeneity in the chemotherapy regimens used, but the different protocols used in the institutions that participated in the study are all based on the same pharmacological backbones. Even if we analyzed the molecular subgroup in almost all patients, risk stratification would still be based mainly on clinical criteria: only recently, studies have begun to show how molecular subgroups, DNA methylation profiles, and next-generation sequencing can complement clinical criteria to better stratify risk, with the potential of molecular tailored therapies [71]. Finally, as already stated, our median follow-up is still relatively short.

## 5. Conclusions

In our experience, PT was associated with a mild acute toxicity profile. Acute and subacute adverse effects were managed with supportive therapy. No treatment interruptions because of radiation-induced toxicity were necessary and no severe acute (G4–G5) adverse effects were observed. Hematological toxicity was also limited, even in the critical cohort of patients who underwent high-dose CSI, which comprises mainly HR patients who receive chemotherapy and, in many cases, autologous HSCT before radiation as well. Preliminary data on late effects are encouraging but need a longer follow-up. There were no cases of post-radiation leukoencephalopathy or brainstem necrosis. Future perspectives will include a thorough examination of late toxicities and treatment response, both of which require longer follow-up.

## Figures and Tables

**Figure 1 cancers-14-02747-f001:**
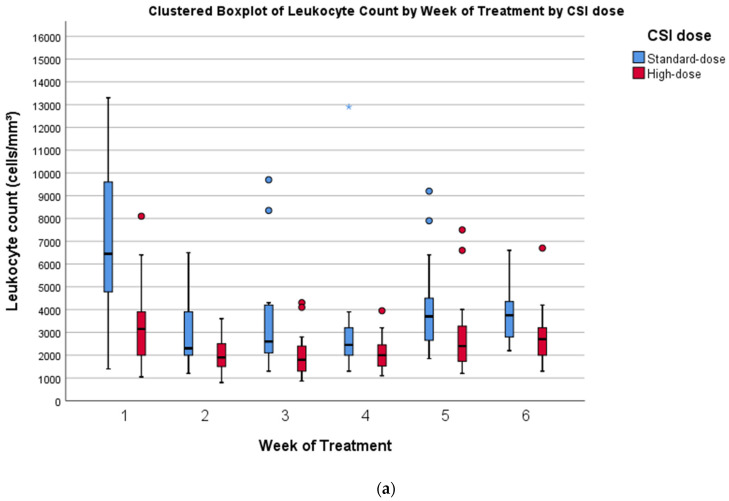
Trends in hematological variables during PT. Clustered boxplots by week of treatment for standard-dose and high-dose CSI: (**a**) leukocyte count (cells/mm³); (**b**) hemoglobin (g/dL); (**c**) platelet count (cells/mm³). Asterisks (blue for standard-dose, red for high-dose) represent extreme outliers.

**Table 1 cancers-14-02747-t001:** Demographic characteristics, pathology, staging, and radiation doses.

Patient Characteristics and Radiation	Total Cohort (*n* = 43)		HR Group (*n* = 26)		SR Group (*n* = 14)		Ex-Infant (*n* = 3)	
Sex	No.	%	No.	%	No.	%	No.	%
Male	23	53.5%	18	69.2%	4	26.7%	1	33.3%
Female	20	46.5%	8	30.8%	10	73.3%	2	66.7%
**Histological subtype**								
Classic	31	72.1%	18	69.2%	12	85.7%	1	33.3%
Desmoplastic	3	7.0%	1	3.8%	1	7.1%	1	33.3%
Large cell/Anaplastic	7	16.3%	7	27.0%	-	-	-	-
Extensive nodularity	1	2.3%	-	-	-	-	1	33.3%
NOS	1	2.3%	-	-	1	7.1%	-	-
**Molecular subgroup**								
Non-WNT/Non-SHH	29	67.4%	21	80.8%	7	50.0%	1	33.3%
SHH	8	18.6%	4	15.4%	2	14.3%	2	66.7%
WNT	3	7.0%	-	-	3	21.4%	-	-
Unavailable	3	7.0%	1	3.8%	2	14.3%	-	-
**MYC amplification**	2	4.7%	2	7.7%	-	-	-	-
**Chang Stage at PT**								
M0	28	65.1%	12	46.2%	14	100.0%	2	66.7%
M1	2	4.7%	2	7.7%	-	-	-	-
M2	2	4.7%	2	7.7%	-	-	-	-
M3	11	25.6%	10	38.4%	-	-	1	33.3%
M4	-	-	-	-	-	-	-	-
	**Median (IQR)**	**Min–Max**	**Median (IQR)**	**Min–Max**	**Median (IQR)**	**Min–Max**	**Median (IQR)**	**Min–Max**
**CSI dose, GyRBE**	36 (12.6)	23.4–39.6	36.0 (0.0)	24.0 *–39.6	23.4 (0.0)	23.4–25.2	36 (6.3)	23.4–36.0
**Total TB dose, GyRBE**	54.0 (0.0)	54.0–55.8	54.0 (0.0)	54.0–55.8	54.0 (0.0)	54.0–55.8	54.0 (0.0)	54.0–54.0
**Boost on mets, GyRBE**	12.6 (9.0)	9.0–18.0	14.4 (9.0)	9.0–18.0	-	-	9.0 (0.0)	9.0–9.0
**Age at diagnosis, years**	8.7 (6.6)	1.8–18.6	10.1 (7.2)	2.5–18.6	6.6 (4.0)	4.0–17.8	1.9 (0.1)	1.8–2.1
**Age at PT start, years**	8.9 (6.8)	3.0–19.0	10.3 (7.3)	3.0–19.0	6.7 (3.9)	4.1–18.0	3.8 (0.1)	3.6–3.8

PT: proton therapy; CSI: craniospinal irradiation; TB: tumor bed; mets = metastasis; *: one high-risk patient was treated with reduced dose CSI (24.0 GyRBE) in accordance with her treatment protocol.

**Table 2 cancers-14-02747-t002:** Surgery, chemotherapy, treatment response, and follow-up duration.

Treatment Characteristics, Response and Follow-Up	Total Sample (*n* = 43)		HR Group (*n* = 26)		SR Group (*n* = 14)		Ex-Infant (*n* = 3)	
Extent of resection								
Partial	20 (46.5%)		15 (57.7%)		2 (14.3%)		3 (100.0%)	
Near-total	2 (4.7%)		-		2 (14.3%)		-	
Gross-total	21 (48.8%)		11 (42.3%)		10 (71.4%)		-	
**HD-CHT with aHSCT prior to PT**	14 (32.6%)		11 (42.3%)		-		3 (100.0%)	
**Concomitant CHT to PT**	18 (41.9%)		13 (50.0%)		5 (35.7%)		-	
**Treatment response**								
CR	35 (81.4%)		20 (76.9%)		12 (85.7%)		3	100.0%
SD	3 (7.0%)		3 (11.5%)		-		-	-
PD	5 (11.6%)		3 (11.5%)		2 (14.2%)		-	-
**Survival at latest FU**								
Alive	41 (95.3%)		24 (92.3%)		14 (100.0%)		3	100.0%
Dead	2 (4.7%)		2 (7.7%)		-		-	-
	**Median (IQR)**	**Min–Max**	**Median (IQR)**	**Min–Max**	**Median (IQR)**	**Min–Max**	**Median (IQR)**	**Min–Max**
**FU duration, months**	26.0 (30.5)	2.0–67.0	20.0 (31.3)	5.0–67.0	28.5 (22.8)	2.0–66.0	24.0 (21.5)	3.0–46.0

HD-CHT: high-dose chemotherapy; aHSCT: autologous hematopoietic stem-cell transplantation; PT: proton therapy; CHT: chemotherapy; CR: complete response; SD: stable disease; PD: progressive disease; FU: follow-up.

**Table 3 cancers-14-02747-t003:** Acute and subacute toxicity.

Type of Toxicity	Total Sample (*n* = 43)		High-Dose CSI (*n* = 27)		Standard-Dose CSI (*n* = 16)	
	G1–G2	G3	G1–G2	G3	G1–G2	G3
**Radiation dermatitis**	28 (65.1%)	1 (2.3%)	17 (63.0%)	1 (3.7%)	11 (68.8%)	-
**Pharyngeal mucositis**	22 (51.2%)	2 (4.7%)	15 (55.6%)	1 (3.7%)	7 (43.8%)	1 (6.3%)
**Nausea/Vomiting**	18 (41.9%)	1 (2.3%)	10 (37.0%)	-	8 (50.0%)	1 (6.3%)
**Alopecia**	17 * (94.4%)	-	3 (100%)	-	14 (93.3%)	-
**Anorexia**	16 (37.2%)	1 (2.3%)	8 (29.6%)	-	8 (50.0%)	1 (6.3%)
**Fatigue**	15 (34.9%)	-	8 (29.6%)	-	7 (43.8%)	-
**Herpes Zoster**	7 (16.3%)	-	4 (14.8%)		3 (18.8%)	
**Headache**	5 (11.6%)	1 (2.3%)	2 (7.4%)	1 (3.7%)	3 (18.8%)	-
**Upper airway infection**	3 (7.0%)	-	2 (7.4%)	-	1 (6.3%)	-
**Insomnia**	2 (4.7%)	-	1 (3.7%)	-	1 (6.3%)	-
**Fever**	3 (7.0%)	-	1 (3.7%)	-	2 (12.5%)	-
**Cough**	1 (2.3%)	-	-	-	1 (6.3%)	-
**Diarrhea**	1 (2.3%)	-	-	-	1 (6.3%)	-
**Myalgia**	1 (2.3%)	-	-	-	1 (6.3%)	-
**PRES**	-	1 (2.3%)	-	1 (3.7%)	-	-
**Cavernoma**	1 (2.3%)	-	1 (3.7%)	-	-	-

*: Percentages of alopecia are calculated considering, in total, only patients without preexisting alopecia (most patients of the high-dose CSI group, due to chemotherapy prior to PT).

**Table 4 cancers-14-02747-t004:** Comparison of mean hematological values between patients who receive high-dose CSI (36.0 GyRBE) and standard-dose CSI (23.4 GyRBE), at different timepoints (weeks of treatment), using the independent samples t-test (Welch’s adaptation was used in the case of unequal variances).

Comparison of High-Dose vs. Standard-Dose CSI	High-Dose CSI (*n* = 27)	Standard-Dose CSI (*n* = 16)	Difference (SE)	95% CI	*p*
Leukocyte Count (Cells/mm³)	Mean (SD)	Mean (SD)			
**Week 1**	3339 (1587)	6679 (3393)	3340 (932)	1382–5298	**<0.00**
**Week 2**	1997 (698)	3118 (1655)	1121 (463)	140–2103	**0.03**
**Week 3**	1972 (879)	3463 (2411)	1492 (648)	119–2865	**0.04**
**Week 4**	2117 (738)	3107 (2804)	990 (738)	−584–2564	0.20
**Week 5**	2814 (1484)	4022 (2157)	1208 (561)	74–2342	**0.04**
**Week 6**	3468 (3641)	3856 (1548)	388 (831)	−1299–2074	0.64
**Hemoglobin value (g/dL)**					
**Week 1**	11.0 (1.0)	11.6 (1.6)	0.5 (0.5)	−0.4–1.5	0.25
**Week 2**	11.1 (1.2)	11.3 (1.6)	0.2 (0.5)	−0.7–1.1	0.63
**Week 3**	11.0 (1.3)	11.2 (1.6)	0.2 (0.5)	−0.7–1.2	0.65
**Week 4**	11.1 (1.0)	11.2 (1.6)	0.1 (0.5)	−0.9–1.1	0.87
**Week 5**	11.2 (1.0)	11.2 (1.7)	0.1 (0.5)	−0.9–1.0	0.92
**Week 6**	11.2 (0.7)	11.2 (1.5)	-0.1 (0.4)	−0.9–0.8	0.92
**Platelet count (cells/mm³)**					
**Week 1**	21,5240 (69,388)	311,300 (103,713)	96,060 (27,333)	40,728–151,392	**<0.00**
**Week 2**	135,590 (41,774)	203,750 (55,424)	68,160 (15,545)	36,691–99,629	**<0.00**
**Week 3**	137,353 (50,476)	192,800 (92,686)	55,447 (25,974)	1083–109,810	**0.04**
**Week 4**	141,436 (54,175)	218,267 (96,355)	76,831 (27,053)	20,209–133,452	**0.01**
**Week 5**	156,385 (54,281)	278,750 (103,357)	122,365 (27,946)	64,070–180,660	**<0.00**
**Week 6**	161,936 (56,438)	276,767 (71,235)	114,831 (20,153)	74,067–155,595	**<0.00**

SD = standard deviation. SE = standard error. CI = confidence interval. Statistically significant results (*p* < 0.05) are in bold.

**Table 5 cancers-14-02747-t005:** Preliminary data on neuroendocrine deficiency.

Therapy	Cases	%
Any non-preexisting HRT	6	14.0
Monotherapy	4	9.3
Multi-drug HRT	2	4.7
**Specific hormones:**		
Thyroxine	5	11.6
Hydrocortisone	3	7.0
Desmopressin	1	2.3
Testosterone	1	2.3
	**Median**	**Min–Max**
Hypothalamic/Pituitary dose, GyRBE	41.24	36.0–55.0
Latency to HRT	9.5 months	3.0–25.0 months

HRT: Hormone Replacement Therapy.

**Table 6 cancers-14-02747-t006:** Preliminary data on ototoxicity.

Patient	CSI/TB dose (GyRBE)	Grade	Laterality	Cochlear D_mean_ (GyRBE)		Previous Chemotherapy
				Right	Left	
1	23.4/30.6	G1	Bilateral	35	28	No
2	36.0/18.0	G2	Bilateral	37	37	Yes
3	36.0/18.0	G2	Bilateral	37	42	Yes
4	23.4/30.6	G2	Bilateral	23	26	No
5	36.0/18.0	G3	Right	53	36	Yes
6	23.4/30.6	G3	Bilateral	32	31	No
7	23.4/30.6	G3	Right	35	42	Yes

Dmean: mean dose.

**Table 7 cancers-14-02747-t007:** Preliminary data on late effects attributable (at least in part) to PT.

Toxicity	High-Dose CSI (*n* = 27)		Standard-Dose CSI (*n* = 16)	
	No. of Cases (%)	Grade	No. of Cases (%)	Grade
Cavernoma	8 (29.6%)	G1	1 (6.3%)	G1
	1 (3.7%)	G2	-	-
Intracranial bleeding	1 (3.7%)	G2	-	-
Loss of visual acuity	-	-	2 (12.5%)	G1
Osteoporosis	1 (3.7%)	G2		
CMV Encephalitis	1 (3.7%)	G4	-	-
Stroke	1 (3.7%)	G1	-	-
RBC Transfusion	1 (3.7%)	G3	-	-
VII CN Paralysis	-	-	1 (6.3%)	G1
Chronic headache	-	-	1 (6.3%)	G1

## Data Availability

Supporting data will be available upon request.

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
