# Peer review of "Toxicity and Clinical Results after Proton Therapy for Pediatric Medulloblastoma: A Multi-Centric Retrospective Study"

_cancers, 2022, doi:10.3390/cancers14112747_

Round 1

Reviewer 1 Report

The article "Toxicity and clinical results after active scanning proton therapy for pediatric medulloblastoma: a multi-centric retrospective study" is well written and concise to the topic. It adds additional information expanding data related to benefits of proton RT in the treatment of pediatric medulloblastoma.

Recommended clarifications:

While the confounding variable of pre-treatment is noted, I believe it to be understated. Given that the volumes and dose per fraction are equivalent until after 13 fraction in CSI SD and HD, these decreased values are attributable to pre-RT chemotherapy commonly given in HR patients. These tables illustrate that fact. Would be more useful to have comparisons within the HR and SR patients treated with whole vertebral body and vertebral body sparing techniques. Any conclusions related to hematologic toxicities need this variable attributed more clearly. (Section 3.2.2 and associated figures)

Similarly, the alopecia data is confusion. Non-preexisting is noted, but the 17 of 43 represents a value against the whole sample without pre-existing patients removed. RT-induced alopecia is likely higher rate with this accounted [17/(43-preexisting)].

CTCAE for hearing loss is not as uniformly known. Was this applied for defining grade of hearing loss, ie threshold shift in dB and Hz? 

The Discussion on RT-induced immune suppression is likely also confounded by pre-RT chemotherapy. Clarification of the 7 cases as related to chemotherapy would be useful. 

Author Response

28th May 2022

Dear Dr. Wang,

Thank you for giving me the opportunity to submit a revised draft of my manuscript titled “Toxicity and clinical results after active scanning proton therapy for pediatric medulloblastoma: a multi-centric retrospective study” submitted to Cancers. We appreciate the time and effort that you and the reviewers have dedicated to providing your valuable feedback on my manuscript. We are grateful to the reviewers for their insightful comments that led to possible improvements in the current version of the manuscript.       
The authors have carefully considered the comments and tried our best to address every one of them. We hope the manuscript after careful revisions meets your high standards. The authors welcome further constructive comments if any.

Below we provide the point-by-point responses. All modifications in the manuscript have been highlighted according to your instruction.

Comments from Reviewer 1

-While the confounding variable of pre-treatment is noted, I believe it to be understated.

Response: Thank you for pointing this out. We agree with this comment. Therefore, we have more clearly stated treatments prior of proton therapy as an important variable to properly evaluate hematological toxicity as following: “The significantly different treatment before PT of HR and SR patients represents however an important variable that should be taken into account for a proper evaluation of hematological toxicity”. This change can be found on page 7, line 256.

-Given that the volumes and dose per fraction are equivalent until after 13 fraction in CSI SD and HD, these decreased values are attributable to pre-RT chemotherapy commonly given in HR patients. These tables illustrate that fact. Would be more useful to have comparisons within the HR and SR patients treated with whole vertebral body and vertebral body sparing techniques. Any conclusions related to hematologic toxicities need this variable attributed more clearly. (Section 3.2.2 and associated figures)

Response: Thank you for this suggestion. It would have been interesting to explore this aspect. However, in the case of our study, it seems slightly difficult, due to the limited number of patients in each subgroup (HR and SR group treated with whole vertebral or vertebral sparing technique) for a comparison. Only 4 patients who received HD CSI were skeletally mature and received the whole dose with vertebral sparing, the other 23 received CSI in two stages as highlighted in the methods. Among SD CSI patients, only 2 were skeletally mature and thus received the whole dose with vertebral sparing. This relevant comment has been highlighted as following: “A comparison between patients treated with different vertebral body irradiation approaches based on skeletal maturity could provide further informative data of hematological toxicity. Nevertheless, such a comparison was not feasible in the present study due to the limited patient number in each subgroup”.  This change can be found on page 12, line 406.

-Similarly, the alopecia data is confusion. Non-preexisting is noted, but the 17 of 43 represents a value against the whole sample without pre-existing patients removed. RT-induced alopecia is likely higher rate with this accounted [17/(43-preexisting)].

Response: We agree with this. A correction of alopecia data has been made, with a clear detail of the number of patients who develop it as a percentage of those without pre-existing alopecia (17 out of 18, 94.4%). This change can be found on page 6, line 216, and the corresponding table was updated.

-CTCAE for hearing loss is not as uniformly known. Was this applied for defining grade of hearing loss, ie threshold shift in dB and Hz?

Response: Thank you for pointing this out. In order to clarify this aspect, we added the requested criteria as following: “Audiologic assessments included bilateral measurement by conventional and extended high frequency audiometry (1, 2, 3, 4, 6 and 8 kHz audiogram). Hearing impairment grading was based on a threshold shift >20 dB at 8, 4, 3 kHz in at least one ear or audiologic indication for cochlear implant (see CTCAE v4.03 for details)”. This change can be found on page 10, line 281.

-The Discussion on RT-induced immune suppression is likely also confounded by pre-RT chemotherapy. Clarification of the 7 cases as related to chemotherapy would be useful.

Response: With the aim of clarify the discussion about immune suppression status in PT patients and development of Herpes Zoster, we stated the absence of correlation to HR vs SR nor a specific treatment prior and during PT in our study as following: “Herpes zoster infection or reactivation was relatively common (7 cases, 16.3%), without a clear correlation with HR or SR group, previous chemotherapy, nor steroid therapy concurrent to PT”. This change can be found on page 6, line 219.

In addition to the above changes, a few spelling errors have been corrected.

We look forward to hearing from you and to respond to any further questions and comments you may have.

Sincerely,

Barbara Rombi, MD PhD
Proton therapy Unit, Santa Chiara Hospital, Azienda Provinciale per i Servizi Sanitari (APSS), 38123, Trento, Italy

Reviewer 2 Report

This is a retrospective outcome analysis of 43 children (median age 8.7) with standard-  and high-risk medulloblastoma (MB) treated with proton therapy  from several centers in Italy and Slovenia.

Overall, important aspects regarding toxicity were systematically worked up.

All limitations, i.e. the retrospective nature of the study, a small number of patients, and the relatively short follow- up period (median 26 months)  are mentioned in the manuscript. Neuropsychological deficits will also only manifest later in the follow-up.

Minor comments:

1.     I would shorten the title to:

Toxicity and clinical results after proton therapy for pediatric medulloblastoma: a multi-centric retrospective study

and mention  “active scanning” only in the text 

2.     Could the authors please explain why standard risk patients received concurrent, single-agent vincristine once a week during irradiation while  high-risk patients received vinorelbine once every two weeks during irradiation ?

3.     Can you already estimate whether proton therapy in children with MB could become a standard in the future taking into account logistical and financial aspects?

Author Response

28th May 2022

Dear Dr. Wang,

Thank you for giving me the opportunity to submit a revised draft of my manuscript titled “Toxicity and clinical results after active scanning proton therapy for pediatric medulloblastoma: a multi-centric retrospective study” submitted to Cancers. We appreciate the time and effort that you and the reviewers have dedicated to providing your valuable feedback on my manuscript. We are grateful to the reviewers for their insightful comments that led to possible improvements in the current version of the manuscript.       
The authors have carefully considered the comments and tried our best to address every one of them. We hope the manuscript after careful revisions meets your high standards. The authors welcome further constructive comments if any.

Below we provide the point-by-point responses. All modifications in the manuscript have been highlighted according to your instruction.

Comments from Reviewer 2

  1. I would shorten the title

Response: Thank you for this suggestion, the title has been modified to “Toxicity and clinical results after proton therapy for pediatric medulloblastoma: a multi-centric retrospective study” This change can be found on page 1, line 2.

  1. Could the authors please explain why standard risk patients received concurrent, single-agent vincristine once a week during irradiation while high-risk patients received vinorelbine once every two weeks during irradiation?

Response: Concerning this comment the explanation of q15 vinorelbine use was guided by a clinical trial built on referring institution experience of concomitant vinorelbine with focal radiotherapy in diffuse pontine glioma (Massimino M et al. J Neurooncol. 2014 Jun;118(2):305-312). With the aims to reduce toxicity (especially myelotoxicity) while maintaining efficacy during CSI, the trial suggested less intense schedule (q15) of standard dose vinorelbine for HR MB patient.  

  1. Can you already estimate whether proton therapy in children with MB could become a standard in the future taking into account logistical and financial aspects?

Response: You have raised an important point here. We tried to estimate more clearly the role of PT for children with MB in the discussion as follows: “Even if available data about clinical efficacy, safety and toxicities of PT in comparison to photon therapy for MB are encouraging, logistical and financial aspects remain an obstacle for a wider use of this technique. Nevertheless, several authors [17–19] believe that it should be strongly considered for MB treatment in children”. This change can be found on page 11, line 339.

In addition to the above changes, a few spelling errors have been corrected.

We look forward to hearing from you and to respond to any further questions and comments you may have.

Sincerely,

Barbara Rombi, MD PhD
Proton therapy Unit, Santa Chiara Hospital, Azienda Provinciale per i Servizi Sanitari (APSS), 38123, Trento, Italy